# Preparation and Boron Removal Performance of Glycidol Modified PANI Nanorods: An Optimization Study Based on Response Surface Methodology

**DOI:** 10.3390/polym15020459

**Published:** 2023-01-15

**Authors:** Yunlong Le, Yunshan Guan, Xiaoying Ma, Weidong Zhang

**Affiliations:** College of Chemical Engineering, Qinghai University, Xining 810016, China

**Keywords:** boron adsorption, adsorbents, polyaniline nanorods, glycidol

## Abstract

Boron removal from aqueous solutions has attracted increasing attention, offering benefits for animal and plant health as well as profound significance for exploiting Salt Lake boron resources. In this work, we synthesized novel glycidol-functionalized and hydrophilic polyaniline (PANI) nanorod adsorbents, which were prepared to separate boron compounds from boric acid aqueous solutions. The as-prepared adsorbents were significantly different from the traditional polymers’ grafting reaction because they had a higher functional yield and more active position for adsorption. The maximum adsorption capacity (0.2210 mmoL∙g^−1^) and optimal adsorption conditions (boric acid concentration of 1307 mg/L, pH = 9.82, time of 10 h) were obtained with single-factor experimentation and the response surface method (RSM). In addition, adsorption kinetics studies showed that the adsorption reaction belonged to the pseudo-first-order kinetic model, and diffusion was the key limiting factor; therefore, the adsorption equilibrium time is more than 10 h. Finally, the related possible adsorption mechanism was investigated based on the species and the diffusion of boron in the aqueous phase.

## 1. Introduction

The research on deboronization is one of the hot points for excess boron in aqueous solutions, which can cause serious health hazards to animals and plants as well as serious damage to the environment [1,2,3,4]. The removal of boron with adsorbents is challenging [5], especially the adsorbents with polyhydric alcohol groups [6,7], on the one hand because of the complex existence of boron in aqueous solutions [8,9,10]; and on the other hand, borate is composed of a variety of boron–oxygen complex anions in aqueous solutions, and its specific composition is affected by concentration, temperature, and pH [11]. Therefore, novel ideal boron adsorbents are urgently necessary with many typical active functional groups (-OH), higher specific surface areas, and multiple recycling (acid and alkali resistant) properties.

With respect to the multiple recycling properties, the acid and alkali resistant properties must be taken into consideration because the desorption process is based on the reaction of adsorption in the presence of a large H^+^ concentration, leading to the adsorption reaction equilibrium moving in the opposite direction [12]. The acid and alkali resistant properties of polymers are very well known, such as polyvinyl chloride, polystyrene, PANI, etc. [8,13,14,15,16]. In addition, the abovementioned polymers not only possess excellent acid and alkali resistant properties but also have reasonable prices. In this paper, we focus on PANI as the raw-materials for the adsorbents based on the inherent advantage: First, the preparation condition is mild, and the corresponding micro-structure can be controlled with in-situ polymerization; second, compared with other polymeric monomers, aniline is cheaper and more widely available; most important of all, the terminal active amino group is a prerequisite for functionalization to adsorbing boron [15,17]. Concerning the adsorbents with polyhydric alcohol groups, the typical ring-opening reaction between -NH_2_ and the epoxy group is a typical functional reaction [18,19], which is different from the traditional grafting reaction with lower grafting yield; moreover, the functional group does not easily fall off from the adsorbents, leading to a decrease in the adsorption capacity as a result of the loss of functional groups.

In this work, PANI-OH-X adsorbents were prepared using in-situ polymerization and the ring-opening reaction between PANI and glycidol for the adsorption of boron in aqueous solutions. First, PANI was prepared with in-situ polymerization. In order to expose more amino groups, the polymerization degree should be lower; simultaneously, the PANI should not be cross-linked with each other. Thus, the micro-structure (nanorods) was controlled by adjusting the reaction conditions. After that, the amino groups that originated from the as-prepared PANI nanorods were activated with the NaOH solution (1 M) treatment. Then the PANI-OH-X adsorbents were prepared with the ring-opening reaction between -NH_2_ (originated from PANI) and glycidol. Afterward, the correlation between the adsorption performance and influential factors of the as-prepared adsorbents (boric acid concentration, pH, and adsorption time) were studied based on single-factor experiments. Finally, an optimization model of the boron adsorption performance for the PANI-OH-X adsorbents was revealed with the response surface method (RSM) as a statistical tool. Moreover, the adsorption mechanism related to the as-prepared PANI-OH-X adsorbents was discussed based on the species and the diffusion of boron in the aqueous phase.

## 2. Materials and Methods

### 2.1. Materials

Aniline (≥99.5 wt%) and glycidol (≥97.0 wt%) were obtained from Shanghai McLean Biochemical Technology Co., Ltd., Shanghai, China. Ammonium persulphate (≥98.0 wt%), sodium hydroxide (≥97 wt%), and perchloric acid (≥70.0 wt%) were purchased from Shanghai Aladdin Biochemical Technology Co., Ltd., Shanghai, China. Boric acid (≥99.5 wt%) was obtained from Tianjin Damao Chemical Reagent Factory, Tianjin, China. Ethanol absolute (≥99.7 wt%) was obtained from Tianjin Fuyu Fine Chemical Co., Ltd., Tianjin, China.

### 2.2. Synthesis of the PANI Nanorods

The PANI nanorods were prepared according to the literature [20]. Typically, 20 mL of aniline was dissolved into 800 mL of a HClO_4_ solution containing 50 mL perchloric acid. Then, the reaction system was stirred and cooled until the system temperature fell below 5 °C. Then, 8.375 g of (NH_4_)S_2_O_8_ was rapidly injected into the above solution, and the reactor was maintained at 0–5 °C with stirring for 24 h. The resulting products were acquired by filtration and washed with distilled water and ethanol several times until the aqueous liquid was colorless.

### 2.3. Synthesis of the PANI Nanorods with Active -NH_2_

The as-prepared dark-blue samples were immersed in 400 mL sodium hydroxide solution (1 M) for 30 min. Then, the slurry was obtained by filtration and immersed in 400 mL fresh sodium hydroxide solution (1 M) for 30 min. Then, the products were prepared by washing with distilled water until the pH values reached approximately 7. After that, in order to remove excess water, the PANI nanorods with active -NH_2_ were washed with 50 mL ethanol 5 times and immersed in 100 mL ethanol for use in the next step.

### 2.4. Preparation of the Glycidol-Functionalized PANI Nanorods (PANI-OH-x)

The as-prepared PANI nanorods with active -NH_2_ and 100 mL ethanol were dispersed in a 250 mL flask. After that, the mixture solution (20 mL ethanol and an amount of the glycidol) was added into the flask dropwise at 85 °C with constant stirring. Then, the reaction continued for 5 h. Finally, the products (glycidol-functionalized PANI nanorods) were obtained by filtering and washing with distilled water for several times and freeze-dried. The final products were collected and stored in a vacuum dryer for the adsorption experiment. For convenience, the dosage of glycidol was 5 mL, 10 mL, 15 mL, and 20 mL, corresponding to PANI-OH-1, PANI-OH-2, PANI-OH-3, and CF@PANI-4, respectively.

### 2.5. Adsorption Experiments

In this study, adsorbent (approximately 0.2000 g) was added to 80 mL of the boric acid solution, then placed into an oscillator at a shaking speed of 210 rpm at 298 K for several hours. The adsorption capacity of the as-prepared PANI-OH-x adsorbents was evaluated with Q (mmoL∙g^−1^), which was calculated according to Formula (1):(1)$$Q=\frac{\left(C_{0}-C_{e}\right)}{10.81\times m}\times V$$
where C_0_ (mg/L) is the initial content of boron, C_e_ (mg/L) is the amount of boron remaining in the solution after adsorption, V (L) is the volume of the boron solution, and m (g) denotes the weight of the PANI-OH-x adsorbents. The concentration of boron in the aqueous solution was determined with the methylamine-h spectrophotometry method using an ultraviolet and visible spectrophotometer. The test standard curve is shown in the Appendix A, and the R^2^ = 0.99987. The corresponding absorbance value (A) was also calculated depending on the concentration of boron (C) as shown in Formula (2):(2)A=0.3988C+0.0059

### 2.6. RSM Experimental Design

To further optimize the adsorption conditions based on the single-experiment data, the response surface methodology was carried out, and the Box–Behnken model was used to investigate the effects of the variables. In this system, the key variables were the adsorption time, boric acid concentration, and pH, which were marked as A, B, and C, respectively. In accordance with the RSM, a total of 17 experimental runs were considered in the empirical model. The main factors and the various experimental design levels considered in this study are presented in Appendix A.

### 2.7. Characterization

The chemical groups of PANI-OH-3 were characterized with Fourier transform infrared spectroscopy (FTIR; SHA-B) at wavelengths of 400 cm^−1^ to 4000 cm^−1^. The elemental composition and valence of the as-prepared PANI-OH-3 were determined with X-ray photoelectron spectroscopy (XPS; ESCALABX), while the micro-morphologies of PANI-OH-3 adsorbents were observed with scanning electron microscopy (SEM, JSM-6610LV).

## 3. Results

### 3.1. Synthesis Mechanism

The related reaction equations to synthesize PANI-OH are summarized in Equations (a), (b), and (c) in Figure 1. First, the PANI nanorods were prepared with in-situ polymerization in the presence of aniline and HClO_4_; in fact, the final products were presented as quaternary ammonium salt with the typical group -NH_3_^+^ ClO_4_^−^, as shown in Equation (a). To realize the ring-opening reaction between the active -NH_2_ and epoxy group, the active -NH_2_ was obtained by the drip washing of the NaOH solution. Therefore, in this paper, the as-prepared PANI nanorods were immersed two times in the 400 mL sodium hydroxide solution (1 M) for 30 min, the reaction Equation (b) occurred, and then washed with distilled water to remove the NaClO_4_. For the reaction Equation (c), the ring-opening reaction between the active -NH_2_ and glycidol is easy to perform without the presence of water in the reaction system. Thus, it was very important when preparing the PANI-OH-x composites that the PANI nanorods with active -NH_2_ compounds were washed with ethanol several times to remove the water. Moreover, the active -NH_2_ is easily oxidized when exposed to the air condition. Therefore, the PANI nanorods with active -NH_2_ compounds must be immersed in ethanol before the reaction Equation (c).

### 3.2. FTIR and XPS Analysis

To illustrate that the -OH groups (glycidol functionalized group) were successfully grafted onto the as-prepared PANI nanorods with active -NH_2_, FTIR was carried out and the results are shown in Figure 2a. A broad peak was observed at 3406 cm^−1^, which was attributed to the stretching vibrations of the -OH group [21], and the peaks at 1600 cm^−1^ and 1500 cm^−1^ were assigned to the bending mode of the -NH_2_ group and -NH- group, respectively [22]. The peaks at 2922 cm^−1^ and 2880 cm^−1^ were assigned to the symmetric and asymmetric stretching of the C-H bands [23], respectively. Compared to the PANI nanorods, a stronger -OH peak (3406 cm^−1^) was displayed at the curve of PANI-OH, indicating that the glycidol successfully reacted with the active -NH_2_, and the glycidol-functionalized group (the red color groups as shown in the image) was successfully synthesized with the ring-opening reaction.

To obtain insight into the elements valence and surface compositions of the as-prepared PANI-OH-3 samples, XPS analysis was carried out to further study the surface composition of the adsorbent before and after boron adsorption, and the results are shown in Figure 2b–h The analysis results showed that PANI-OH-3 and PANI-OH-3-B (the adsorbents after adsorption of boron compounds) contained C, O, and N elements (Figure 2b). For PANI-OH-3-B, the XPS C 1s spectrum (Figure 2c) was curve-fitted to four peaks with binding energies of 284.25 eV, 284.8 eV, 285.45 eV, and 286.4 eV, which were attributed to C-H, C-C, C-N, and C-O [24,25,26], respectively; moreover, the C 1s regions of PANI-OH-3 and PANI-OH-3-B were without obvious changes. Concerning the O 1s regions of PANI-OH-3 and PANI-OH-3-B, for PANI-OH-3, the dominant peak is located at 532.8 eV, which was attributed to the C-O that originated from C-OH (Figure 2d), while the two peaks with binding energies of 532.5 eV and 533.1 eV were assigned to C-O and B-O [27,28] as a result of the presence of C-OH and C-O-B (Figure 2e). The binding energies between 399.29 eV and 399.7 eV consisted of the N1s peak (Figure 2f), which was ascribed to -N = and -NH- [24,26], respectively. In addition, the N 1s regions of PANI-OH-3 and PANI-OH-3-B were without obvious changes. For Figure 2g, the Cl 2p peaks of PANI appearing at 207.85 eV and 209.4 eV can be assigned to the ClO_4_^−^ [29,30]. Comparing the overall XPS images before and after adsorption (PANI-OH-3 and PANI-OH-3-B), the B 1s peaks were located at 191.99 eV, which was attributed to B-O [31], indicating that boron was successfully adsorbed with the PANI-OH-3 adsorbent (Figure 2g).

### 3.3. SEM Analysis

The microstructures of the as-prepared PANI-OH-x adsorbents were significantly correlated to their adsorption capacity. Therefore, SEM analysis was carried out and shown in Figure 3. The as-prepared PANI-OH-x adsorbent powders were composed of the PANI nanorods cluster, which looks similar to the carbon nanotube cluster (Figure 3a). In addition, concerning the amplified image (Figure 3b), the length of the as-prepared PANI nanorods was approximately 600 nm, and the diameter was approximately 40 nm. Thus, it can be inferred that the length of the as-prepared PANI nanorods significantly affected the proportion of the efficient glycidol-functionalized group. The SEM and corresponding EDS mapping images were investigated and shown in Figure 3c; obviously, the elements of C and N were uniformly distributed around the PANI nanorods’ skeleton.

### 3.4. Discussion of the Adsorption Experiments

#### 3.4.1. Optimization of Adsorbents

For the adsorbents, the proportion of the efficient glycidol-functionalized group, which is the glycidol addition, played an important role in the boron adsorption capacity. As shown in Figure 4a, the adsorption capacity gradually increased with the addition of glycidol. The adsorption capacity reached a maximum value (0.1013 mmoL∙g^−1^) when the addition of glycidol was 20 mL. Moreover, the adsorption capacity was 0.0512 mmoL∙g^−1^, 0.0924 mmoL∙g^−1^, and 0.1006 mmoL∙g^−1^ corresponding to the addition of 5 mL, 10 mL, and 15 mL glycidol, respectively. From what has been discussed above, it can be inferred that, with an addition of glycidol up to 15 mL, the ring-opening reaction reaches the maximum degree. Excess glycidol added to the PANI nanorods could not be fixed and could not take part in the ring-opening reaction. Therefore, the PANI-OH-3 was considered the best adsorbent for this adsorption system.

#### 3.4.2. Single-Factor Experiment

According to the previous literature and Equation (3),
(3)$$\begin{array}{c} B{\left(\mathrm{OH}\right)}_{3}+H_{2}O⇌B{\left(\mathrm{OH}\right)}_{4}{}^{-}+H^{+}\\ K_{a}=10^{-9.2} \end{array}$$
the species of boron in the boric acid solution of this adsorption system were investigated, and the results are shown in Appendix A based on the pH value and concentration of initial boric acid as well as weak boric acid with a pK_a_ value of 9.2. Therefore, the pH value of the solution was significant for chemical adsorption, especially based on the reaction of Equational (3). Figure 4b shows the adsorption capacity dependent on the pH values. When the pH value was approximately 10, the maximum adsorption capacity was 0.113 mmoL∙g^−1^. In addition, when the pH value was approximately 7, adsorption was very weak because of the boron form in the aqueous solution, where the dominant species was B(OH)_3_, which was not beneficial for realizing the adsorption reaction. On the other side, when the pH value was approximately 8–10, the maximum adsorption capacity significantly increased, which originated from the equilibrium of the adsorption reaction moving forward because the hydrogen ions produced with the adsorption reaction were neutralized with the alkaline reaction system However, when the pH of the solution was greater than 10, the decrease in the boron adsorption capacity was attributed to the repulsive interactions between the deprotonated hydroxyl groups of the polyhydric alcohols that originated from glycidol and the borate anions as well as to the competition between the borate and hydroxide anions for the adsorption sites.

Figure 4c shows the correlation between the boron initial concentration and the adsorption capacity of the PANI-OH-3. As illustrated in the figure (in the Appendix A), the adsorption capacity was highest when the initial boron concentration was 1300 mg/L, indicating the adsorption process was favorable at high boron concentrations, which was also in accordance with the equilibrium of the adsorption reaction moving forward due to the higher initial reaction concentration.

Concerning the adsorption time, the adsorbent for this system could be assigned to chemical adsorption; thus, the reaction time was significantly correlated with the adsorption capacity. The effect of time on the adsorption capacity was investigated at 298 K, and the results are shown in Figure 4d. The adsorption capacity significantly increased when the reaction time was less than 6 h, while the maximum adsorption capacity was reached when the reaction time was approximately 12 h. This was because the initial boron solution concentration was higher and the adsorbent had more active sites; thus, the adsorption rate was faster. As boron continued to occupy the adsorption sites until all of the adsorption sites were filled, the adsorption capacity became saturated and reached the maximum adsorption capacity. On the other hand, the adsorption is a chemical reaction; thus, there were collisions between the boron compounds in the aqueous phase and polyhydric alcohols that originated from glycidol. Therefore, we can infer that the diffusion of the boron compounds in the aqueous phase is one of the key limiting factors for boron removal.

The relationship of adsorption capacity and adsorption time was revealed with the adsorption kinetics based on the single experimental data, where the pseudo-first-order and pseudo-second-order models were used for interpreting the phenomenon.

The pseudo-first-order equation is shown in Formula (4):(4)$$\ln\left(q_{e}-q_{t}\right)=\ln q_{e}-k_{1}t$$
where q_t_ (mg/g) is the amount of boron adsorbed at time t, q_e_ (mg/g) is the maximum adsorption capacity for the pseudo-first-order adsorption, and k_1_ is the pseudo-first-order rate constant for the boron adsorption process (min^−1^).

The pseudo-second-order equation is shown in Formula (5):(5)$$\frac{t}{q_{t}}=\frac{1}{k_{2}\times q_{e}{}^{2}}+\frac{t}{q_{e}}$$
where q_t_ (mg/g) is the amount of boron adsorbed at time t, q_2_ (mg/g) is the maximum adsorption capacity for the pseudo-second-order adsorption, and k_2_ is the pseudo-second-order rate constant (g·mg^−1^·min^−1^).

As shown in Appendix A, according to the fitting results, the pseudo-first-order kinetic model was more suitable because the value of R^2^ was 0.995, which meant that the theoretical equilibrium values obtained from the pseudo-first-order kinetic model were closer to the experimental values. With the progression of the adsorption reaction, the remaining concentration of the PANI-OH-3 active groups remained basically unchanged. For this adsorption system, k_1_ was only 0.2848, suggesting that diffusion was the key limiting factor to the reaction (Appendix A); however, the value of R^2^ was 0.949391, which can be assigned to the shock treatment not completely static adsorption. In any case, the diffusion is one of the key limiting factors for this system’s adsorption.

#### 3.4.3. RSM Experimental

To further optimize the adsorption condition, the response surface method was carried out in this paper. The ANOVA test data for the response surface Quadratic model analysis are presented in Table 1, which used Design-Expert.V8.0.6.1. When evaluating the ANOVA table, the parameter with the highest F value (490.77) and lowest *p* (probability) value (<0.0001) was found to be the boric acid concentration. The model’s F-value of 6799.12 implied that the model was significant, and the determination coefficient values between the experiment and model were R^2^ = 0.9999, adjusted-R^2^ = 0.9997, and predicted-R^2^ = 0.9988. These metrics showed that the model was highly compatible with the experimental data. In addition, the lack of fit value was 0.2098 (*p* > 0.05). Therefore, the predictive models derived from the RSM predicted the experimental data well. The corresponding adsorption capacity (Q_e_) was also calculated depending on the RSM, as shown in Formula (6):(6)$$\begin{array}{l} Q_{e}=-2.17357+0.028595A+0.00208798B+0.177954C-3.25\times 10^{-6}\mathrm{AB}-\\ 0.0004\mathrm{AC}-4.725\times 10^{-6}\mathrm{BC}-0.000815A^{2}-7.68875\times 10^{-7}B^{2}-0.008539\times C^{2} \end{array}$$

The desirability of R^2^ = 0.9999 confirmed the acceptability and applicability of the model, indicating that the RSM model was a useful technique for optimizing conditional designs. In addition, it was in good agreement with the predicted results, which confirmed the adequacy and effectiveness of the model. Using the optimization analysis section of Design-Expert.V8.0.6.1, the optimal conditions were predicted as follows: boric acid concentration of 1306.57 mg/L, pH of 9.82, time of 10 h, and a maximum adsorption capacity of 0.228078 mmoL∙g^−1^. To examine the accuracy and confidence of the proposed optimization conditions, the experiments were carried out at the abovementioned conditions for five groups, except the real boric acid concentration was 1306 mg/L. The measured average adsorption capacity value was 0.2271 mmoL∙g^−1^, which proves that the response surface optimization model is highly compatible and can be used to optimize the experimental conditions for this system.

The results of the interactions between the three different factors based on the 3D surface plots are presented in Figure 5, and Figure 5a–c shows the adsorption capacity with the different pH values at a fixed condition, when the reaction time and initial boron concentration were10 h and 1300 mg/L, respectively. The adsorption capacity at a pH of 10 was much higher than when the pH was equal to 9 and 11. In addition, at a pH of 9, the corresponding maximum adsorption capacity was only 0.2224 mmoL∙g^−1^. At a pH of 10, the corresponding maximum adsorption capacity reached 0.228078 mmoL∙g^−1^. However, when the pH reached up to 11, the corresponding maximum adsorption capacity decreased to 0.1971 mmoL∙g^−1^. The adsorption capacity was closely related to the initial boron concentration as shown in Figure 5d–f, which can be assigned to the initial boron concentration acting as one of the reactants and greatly affecting the forward shift degree of the reaction equilibrium. However, too much boric acid and the adsorption capacity will go down, as a result of the species of boron compounds not benefiting the typical adsorption reaction. With respect to the adsorption time (Figure 5g–i), the adsorption capacity is significantly dependent on the reaction time; the dominant reason can be assigned to the diffusion of boron compounds in the aqueous phase. Therefore, the pH and time were the more important limiting factors for the adsorption capacity compared to the time, initial boron concentration, and pH. These observations coincided well with the single-experiment discussion.

#### 3.4.4. The Regeneration of the Synthesized Adsorbent and Their Inherent Properties

A total of 1.200 g of the PANI-OH-3 adsorbents after adsorption were immersed into 600 mL HCL solution (0.5 M) then placed into an oscillator at a shaking speed of 210 rpm at 298 K for 10 h. After that, the adsorbents were filtered and washed several times with distilled water until the filtrate pH value reached approximately 7; then, the adsorbents were obtained by freeze-drying for the next adsorption experiments. The adsorption capacity (q_n_) of the adsorbents after several time regenerations was investigated, and the results are shown in Figure 6. Obviously, the maximum adsorption capacity of PANI-OH-3 decreased approximately 12.57% after 5-times regeneration, which can be assigned to incomplete desorption of adsorption or the little loss of adsorbents due to the manual filtering operation. In any case, the maximum adsorption capacity of PANI-OH-3 decreased approximately 5%, while the regeneration time was less than 3; thus, this adsorbent is stable and has reusable characteristics.

On the other hand, according to the decrease in the maximum adsorption capacity of the regenerated adsorbents, we can infer that the incomplete desorption looks impossible because the incomplete detachment ratio is a stable coefficient not an increased value for a fixed desorption reaction. Thus, the loss of maximum adsorption capacity originated from the loss of adsorbents due to the manual filtering operation.

To further reveal the advantages and disadvantages of the as-prepared adsorbents, the related adsorbents were compared, and the results are shown in Table 2. It can be found that the as-prepared adsorbents’ maximum adsorption capacity is approximately 3-times that of commercial adsorbents (Amberlite IRA743), which indicates that the as-prepared adsorbent possesses excellent adsorption and reusable properties. Concerning the adsorption condition, we can infer that this adsorbent can be used in a strong alkaline environment, where the pH value is approximately 10, at which other adsorbents cannot be used. In the practice application, the brine of Salt Lake is approximately 10; thus, it is very important for the practice to directly remove boron from brine using this type of adsorbent. However, the disadvantages of PANI-OH-3 are the as-prepared adsorbents were powder with low density, the powder adsorbents cannot be directly used in the practice application, and the powder adsorbents must be further treated by prilling combined with adhesive. However, the prilling process will significantly decrease the adsorption capacity.

### 3.5. Adsorption Mechanism

To further interpret the possible adsorption mechanism of this adsorption reaction, several factors and phenomena attracted our attention. For this adsorbent, the adsorption is a type of complexation, and the typical reaction could be simply described as Figure 7. The boron species ([B(OH)_3_] or [B(OH)_4_^−^]) in the boric acid solution was one of the important reactants for boron adsorption (as shown in Appendix A). These were found to not ionize and exist mainly in the form of [B(OH)_3_], with a pH value of less than or equal to 7. However, if the pH exceeds 9, the predominant species of boron in the boric acid solution is [B(OH)_4_^−^]. In this study, the optimal pH for the adsorption of boron was 9.82, and the species of boron in this system was [B(OH)_4_^−^] with a concentration of up to approximately 90%, suggesting that the dominant species of boron involved in the adsorption reaction was [B(OH)_4_^−^] as a result of the weak acid B(OH)_3_. Thus, we can infer that approximately 90% of the boron was adsorbed with the reaction (b) as the result of the maximum adsorption capacity carried out with the pH value approximately 10 (the boron species in the boric acid solution was 90% [B(OH)_4_^−^]). Thus, we can conclude that the dominant adsorption reaction originated from the reaction between [B(OH)_4_^−^] and the alcohol groups for the adsorbents with polyhydric alcohol groups. In addition, the species of boron in the aqueous phase is closely related to the pH values and the initial boron concentration. Hence, the species of boron in the aqueous phase is most important and a key limiting factor to the boron adsorption, especially the adsorbents with polyhydric alcohol groups.

Concerning the isotherm models, there are two kinds of typical isotherm models: Langmuir and Freundlich [35,36]; these were taken into consideration, and the related models’ equations are shown in Table 3, where the *K_L_*, *K_F_*, *Q_m_*, *Q_e_*, and *C_e_* represent the coefficient of Langmuir, the coefficient of Freundlich, the maximum adsorption capacity, equilibrium adsorption capacity, and equilibrium concentration of boron, respectively. The experimental conditions can be briefly described as pH = 10, adsorption temperature: 25 °C, initial boron concentration from 100 mg/L to 1500 mg/L, and the adsorption time approximately 10 h. Finally, the results are shown in Table 3 and Figure 8. With respect to the values of R^2^, the Langmuir was higher than that of Freundlich; thus, the experimental values were more consistent with the Langmuir model. It can be concluded that the adsorption reaction is chemical adsorption, and the ratio between the adsorbents (PANI-OH-3) and boric acid is 1:1; the theoretical maximum adsorption capacity can be up to 0.35682 mmoL∙g^−1^.

## 4. Conclusions

In this study, according to SEM, FTIR, and XPS analyses, the PANI-OH-3 adsorbents were successfully prepared with in-situ polymerization and the ring-opening reaction. The optimal adsorption conditions (boric acid concentration of 1306.57 mg/L, pH of 9.82, and time of 10 h) were obtained based on the single-factor experiment and the RSM method, and the maximum adsorption capacity was 0.228078 mmoL∙g^−1^; for the regeneration, their predominately decreased maximum adsorption capacity is attributed to the manufacture filtering operation. The related adsorption mechanism based on the as-prepared PANI-OH-3 adsorbents can infer the following:(1)The species of boron in the aqueous phase is most important and a key limiting factor to the boron adsorption reaction, especially the adsorbents with polyhydric alcohol groups, as a result of the reaction between the PANI-OH-3 adsorbents and [B(OH)_4_^−^] that accounts for 90% of the total adsorption reaction;(2)The dominant isotherm models coincide with the Langmuir model, which predicted the adsorption reaction is a chemical adsorption, and the ratio between the adsorbents (PANI-OH-3) and boric acid is 1:1; the theoretical maximum adsorption capacity can be up to 0.35682 mmoL∙g^−1^.

## Figures and Tables

**Figure 1 polymers-15-00459-f001:**
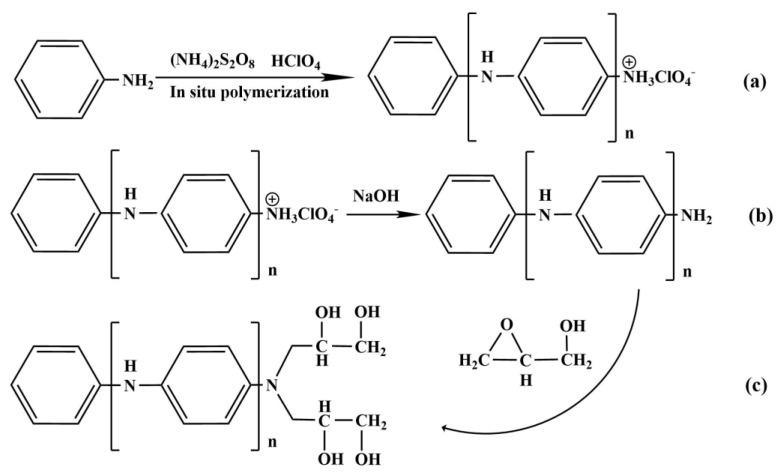
The typical reaction mechanism for the PANI-OH-3 adsorbents. (**a**) In situ polymerization of aniline; (**b**) reducing activity -NH_2_; (**c**) graft functional group.

**Figure 2 polymers-15-00459-f002:**
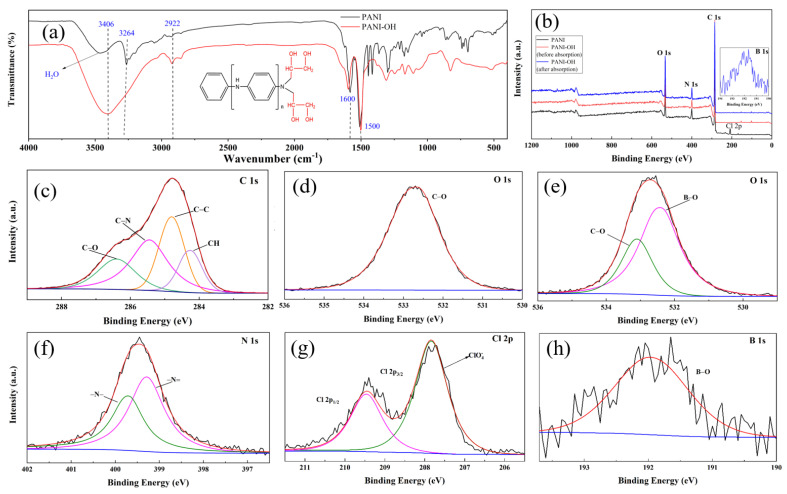
FTIR patterns of the as-prepared PANI and PANI-OH-3 composites (**a**) and the XPS spectra of the PANI and PANI-OH-3 as well as PANI-OH-3-B composite: experimental and fitted curves, (**b**) Full-scan XPS spectra of PANI, PANI-OH-3 and PANI-OH-3 after adsorption, (**c**) C 1s of PANI-OH-3, (**d**) O 1s of PANI-OH-3, (**e**) O 1s of PANI-OH-3-B, (**f**) N 1s of PANI-OH-3, (**g**) Cl 2p of PANI, and (**h**) Fe 2p of PANI-OH-3-B regions.

**Figure 3 polymers-15-00459-f003:**
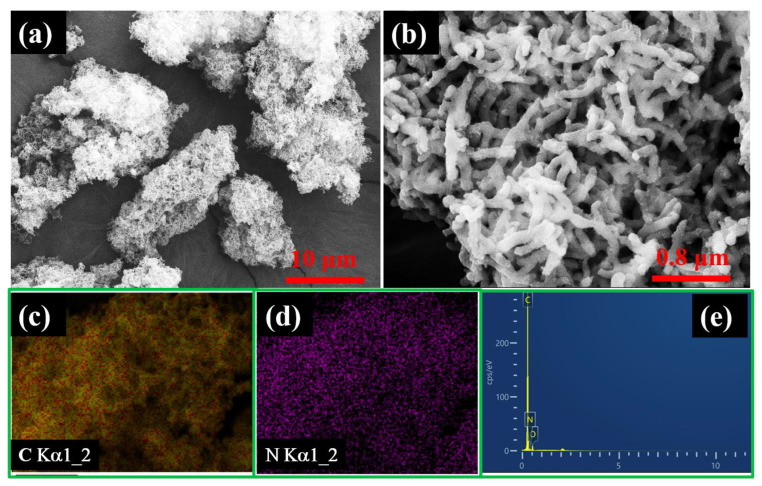
SEM images of the as-prepared PANI-OH-3 composites with different sizes (**a**,**b**) and the corresponding EDS elements mapping (**c**–**e**).

**Figure 4 polymers-15-00459-f004:**
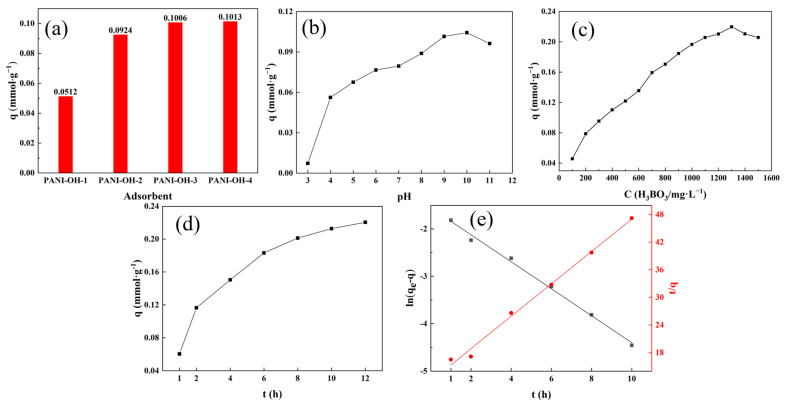
The effect of (**a**) glycidol content (pH = 9, C_0_ = 600 mg/L, t = 10 h, T = 298 K); (**b**) pH (C_0_ = 600 mg/L, t = 10 h, T = 298 K); (**c**) C_0_ (pH = 10, t = 10 h, T = 298 K); (**d**) t (pH = 10, C_0_ = 1300 mg/L, T = 298 K) on the adsorption capacity; (**e**) adsorption kinetics fitting.

**Figure 5 polymers-15-00459-f005:**
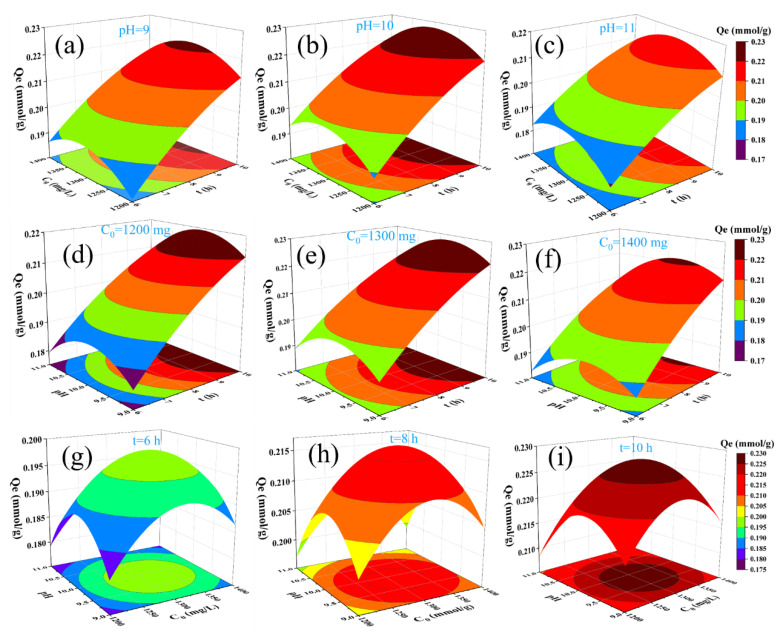
Binary interactions of (**a**) t and C_0_ (pH = 9); (**b**) t and C_0_ (pH = 10); (**c**) t and C_0_ (pH = 11); (**d**) pH and t (C_0_ = 1200 mg); (**e**) pH and t (C_0_ = 1300 mg); (**f**) pH and t (C_0_ = 1400 mg); (**g**) pH and C_0_ (t = 6 h); (**h**) pH and C_0_ (t = 8 h); and (**i**) pH and C_0_ (t = 10 h) on the adsorption capacity of boric acid.

**Figure 6 polymers-15-00459-f006:**
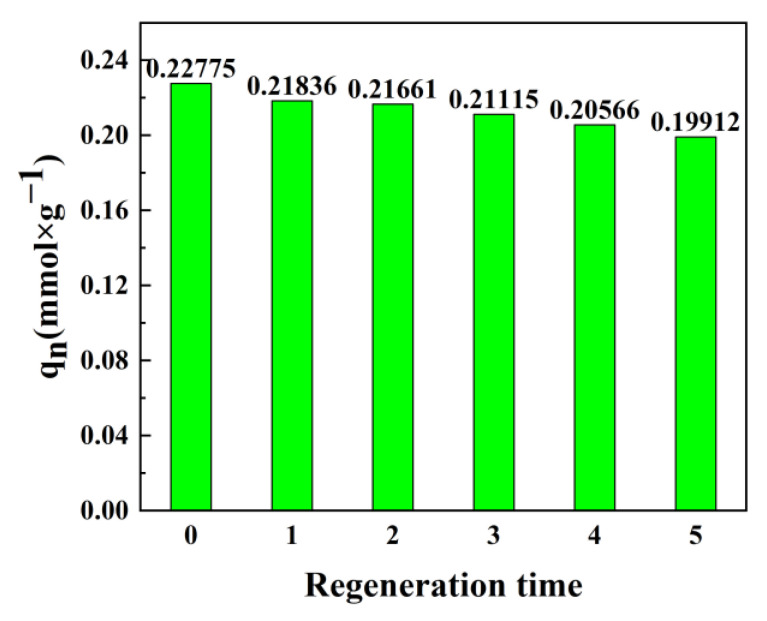
The adsorption capacity of the adsorbents after several times regeneration.

**Figure 7 polymers-15-00459-f007:**
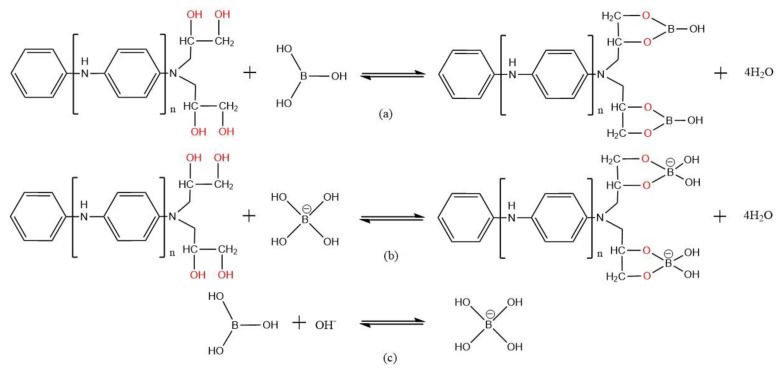
The possible adsorption mechanism for the as-prepared PANI-OH-3 adsorbents. (**a**) Complexation reaction between adsorbent and B(OH)_3_; (**b**) Complexation reaction between adsorbent and [B(OH)4^−^]; (**a**,**c**) Hydrolysis of B(OH)_3_ to [B(OH)4^−^].

**Figure 8 polymers-15-00459-f008:**
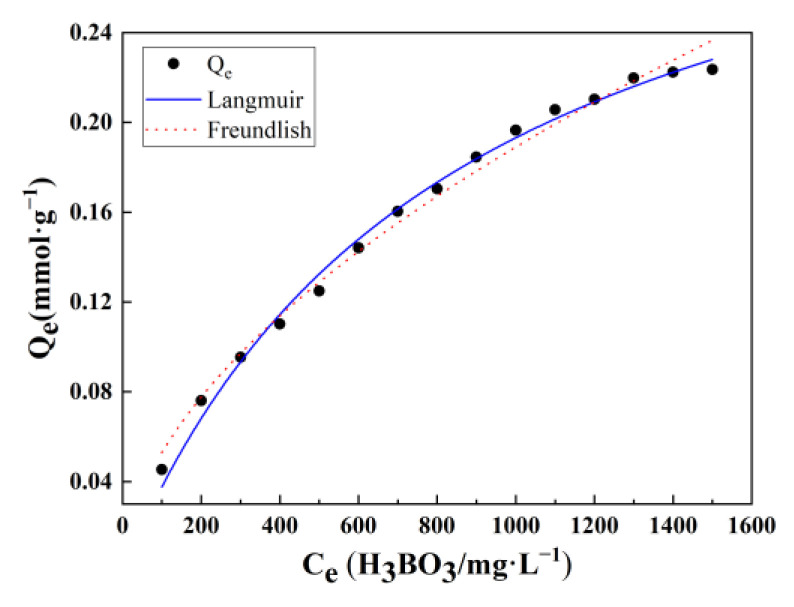
The comparison of the experimental values with the simulated values of Langmuir and Freundlich for the as-prepared PANI-OH-3 adsorbents.

**Table 1 polymers-15-00459-t001:** ANOVA table for the boric acid adsorption capacity on the meglumine-functionalized modified graphene oxide/carbon nanotube aerogels.

Source	SS	df	MS	F-Value	*p*-Value	
Modle	0.0024	9	0.0003	6799.12	<0.0001	significant
A-time	0.0017	1	0.0017	43,074.49	<0.0001	
B-C	0.0000	1	0.0000	490.77	<0.0001	
C-pH	0.0000	1	0.0000	936.66	<0.0001	
AB	1.690 × 10^−6^	1	1.690 × 10^−6^	42.40	0.0003	
AC	2.560 × 10^−6^	1	2.560 × 10^−6^	64.22	<0.0001	
BC	8.930 × 10^−7^	1	8.930 × 10^−7^	22.40	0.0021	
A^2^	0.0000	1	0.0000	1123.46	<0.0001	
B^2^	0.0002	1	0.0002	6244.57	<0.0001	
C^2^	0.0003	1	0.0003	7701.58	<0.0001	
Residual	2.790 × 10^−7^	7	3.986 × 10^−8^			
Lack of Fit	1.790 × 10^−7^	3	5.967 × 10^−8^	2.39	0.2098	not significant
Pure Error	1.000 × 10^−7^	4	2.500 × 10^−8^			
Cor Total	0.0024	16				
C.V.% = 9.63%	R^2^ = 99.99%	Adjust R^2^ = 99.97%		Predicted R^2^ = 99.88%		

**Table 2 polymers-15-00459-t002:** Comparisons of the maximum adsorption capacity between our work and related reports in the literature.

Adsorbent	T (°C)	C_0_ (mg/L)	pH	q_max_ (mmol/g)	References
PE/PP-g-PVAm-G	78.9	171.6	7	0.415	[32]
CQDs/LDHs	45	143	8.5	0.315	[33]
ATG	30	915	8.8	0.11	[34]
Amberlite IRA743	25	28.6	8	0.087	[1]
PANI-OH	25	1307	9.82	0.2281	This works

**Table 3 polymers-15-00459-t003:** Fitting the parameters of the Langmuir and Freundlich equations to the PANI-OH-3 adsorbents.

Models and Parameters	Langmuir			Freundlich		
	$Q_{e}=\frac{Q_{m}K_{L}C_{e}}{1+K_{L}C_{e}}$			$Q_{e}=K_{F}*C_{e}{}^{\frac{1}{n}}$		
	*K_L_*	*Q_m_*	R^2^	*K_F_*	*n*	R^2^
value	0.00118	0.35682	0.99376	0.00415	1.80849	0.98924

## Data Availability

Not applicable.

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
