# Peer review of "Preparation and Boron Removal Performance of Glycidol Modified PANI Nanorods: An Optimization Study Based on Response Surface Methodology"

_polymers, 2023, doi:10.3390/polym15020459_

Round 1
Reviewer 1 Report
The study is good, but it has many flaws that have not been investigated and the study is incomplete.
Why isotherm models were not used in the study?
Why are thermodynamic studies not used in the study?
The studied adsorbent should be compared with other adsorbents in terms of adsorption side, which has not been done
Units in all sections should be uniform, significant figures and tables should be uniform.
Improve the conclusion to deliver more qualitative information.
The novelty of this research should be inserted in the text clearly.
The advantages and disadvantages of the synthesized adsorbent should be investigated.
The ion leaching from the adsorbent during the degradation process should be investigated.
The regeneration of the synthesized adsorbent should be studied.
Authors need to go through all result seriously and need to discuss them accordingly.
Author Response
Please check the attachment, thank you very much for your works.

Reviewer 2 Report
The work is interesting and useful, too. However, the following are the comments.
(i) The PANI nanorods were prepared according to the literature [20].Provide a brief description of PANI nanorods from the literature [20].
(ii) On 2.3. Synthesis of the PANI nanorods with active -NH2: the as-prepared dark blue samples were immersed in the 400 mL sodium hydroxide 80 solution (1M) for 30 min. The word from starting sentence should be capital.
(iii) The reviewer noticed several portions of the manuscript are copied from the literature [Langmuir 2022, 38, 48, 14879–14890]. Reduce similarity index.
(iv) The quality of figure 2 should be improved.
(v) On the caption of Figure 3. SEM images of as-prepared of PANI-OH-3 composites with different size ((a) and (b)) and the corresponding EDS elements mapping. There were not included Figure 3(c) and rest images. It looks more images on Figure 3. The authors mentioned up to Figure 3(c) and there are other images, too. Make it clear.
(vi) English language needs to correct through out the manuscript.
(vii) The reviewer suggested to the authors to use own language through out the manuscript.
(viii) The equation (3-4) is not cleared.
(ix) Rewrite the conclusion by providing exact findings from results and discussion. The reviewer noticed incomplete sentence as (2) the diffusion of boron compounds in the aqueous phase is one of the keys limiting factors for boron remove, which is significantly related to the adsorption time;

Author Response
Thank you very much for your kind works, please check it in the revised manuscript.

Round 2
Reviewer 1 Report
accept
Reviewer 2 Report
The revised manuscript looks good.